# Nurturing Practitioner-Researcher Partnerships to Improve Adoption and Delivery of Research-Based Social and Public Health Services Worldwide

**DOI:** 10.3390/ijerph16050862

**Published:** 2019-03-09

**Authors:** Rogério M. Pinto, Anya Y. Spector, Rahbel Rahman

**Affiliations:** 1University of Michigan School of Social Work—Room 2850, 1080 South University, Ann Arbor, MI 48109, USA; 2Stella and Charles Guttman Community College, City University of New York, 50 West 40th Street, New York, NY 10018, USA; anya.spector@guttman.cuny.edu; 3Graduate School of Social Service, Fordham University Room 721-F, 113 West 60th Street, New York, NY 10023, USA; rrahman20@fordham.edu

**Keywords:** community-engaged research, international participatory research, research to practice, research-based interventions, practitioner–researcher partnership

## Abstract

Research-based practices—psychosocial, behavioral, and public health interventions—have been demonstrated to be effective and often cost-saving treatments, but they can take up to two decades to reach practitioners within the health and human services workforce worldwide. Practitioners often rely on anecdotal evidence and their “practice wisdom” rather than on research, and may thus unintentionally provide less effective or ineffective services. Worldwide, community engagement in research is recommended, particularly in low-resource contexts. However, practitioner involvement has not been adequately explored in its own right as an innovative community-engaged practice that requires a tailored approach. The involvement of practitioners in research has been shown to improve their use of research-based interventions, and thus the quality of care and client outcomes. Nevertheless, the literature is lacking specificity about when and how (that is, using which tasks and procedures) to nurture and develop practitioner–researcher partnerships. This paper offers theoretical and empirical evidence on practitioner–researcher partnerships as an innovation with potential to enhance each phase of the research cycle and improve services, using data from the United States, Brazil, and Spain. Recommendations for partnership development and sustainability are offered, and a case is made for involving practitioners in research in order to advance social justice by amplifying the local relevance of research, increasing the likelihood of dissemination to community settings, and securing the sustainability of research-based interventions in practice settings.

## 1. Introduction

Social work, public health, and medical research findings can take up to two decades to reach and influence the health and human services workforce, including physicians, nurses, social workers, health educators, community-health workers and other practitioners [1,2,3]. This gap exists for a myriad of reasons including dissemination issues, issues in applicability of research findings to practice, and the limitations of a top-down approach that favors researchers’ opinions and expertise, rather than prioritizing bi-directional communication between practitioners, policy-makers, and researchers. This workforce has an ethical obligation to offer up-to-date, evidence-based interventions (EBIs) shown to improve health outcomes [4,5,6]. EBIs are limited to experimental and quantitative designs—most commonly, randomized controlled trials—and are most often associated with medicine or clinical care [7]. More broadly, research-based practices are grounded in qualitative and quantitative methods: Case studies, ethnographies, and quasi-experimental or experimental studies. Both EBIs and research-based practices are useful to practitioners offering health, social, mental, vocational, and developmental services. However, most research findings applied by practitioners in their day-to-day work do not originate in clinical trials. Rather, practitioners may be relying on social and public health research that is focused on understanding social and contextual issues, challenges, and perspectives—research that is grounded in qualitative interviews, observational studies, focus groups, and surveys [8,9].

Well-intentioned practitioners often perceive research as complicated; they may find it difficult to understand top-down impositions mandated by administrators [10]. Practitioners often opt to provide services based on “practice wisdom” (anecdotal “best practices” and professional and personal experiences) instead of scientific evidence [11,12,13,14]. Consequently, clients who seek services at health and human services organizations may receive outdated interventions of unknown quality [15,16]. This can result in additional costs and time spent, and even low levels of satisfaction with services on the part of clients who may become frustrated that they are not experiencing the expected benefits of treatment [17].

Research shows that practitioners personally involved in health and social science research are more likely to buy into and to offer research-based services [18,19,20,21,22]. Thus, community-engaged research involving practitioners has the potential to bridge research and practice and improve outcomes for a variety of health and social needs [23,24,25,26]. Though the importance of collaboration between researchers and community residents has been emphasized in the literature [27], practitioners have been largely overlooked as collaborators who can provide research questions, advance agendas, inform priorities, and interpret research findings [28]. Furthermore, practitioners have, in addition to their ethical mandate to provide high-quality, research-based treatment, an ethical obligation to advance social justice through engagement with and advocacy for individuals, communities, and populations that have been disadvantaged, underserved, or oppressed [29]. As community-engaged research and implementation of research-based practices proliferate, practitioner–researcher partnerships will face challenges. Researchers favoring full control over the research may be unable to sustain partnerships. Cultural differences between the service and academic spheres may also hinder and/or facilitate partnerships.

This paper offers a theoretically and empirically based characterization of practitioner engagement in research and provides strategies to improve practitioners’ use of research to guide practice. Practitioners are central to adopting, implementing, and sustaining the use of research in practice [30]. Herein we provide both theoretical and empirical support for practitioner–researcher partnerships as innovations—new ideas, programs, or technologies based on research [31]. Innovations can be implemented across systems of health and human services organizations. Practitioners can help ensure that research-based practices will be disseminated and adopted more efficiently [32,33]. Practitioners are encouraged to adopt and implement practices that have been shown to be effective in promoting health and preventing disease through different research methods, such as qualitative, experimental, and quasi-experimental studies [34,35,36]. In this paper, we explain the tasks and procedures performed by practitioners involved in research. By illustrating the procedures and tasks performed by practitioners in research collaborations, we affirmed the idea that such partnerships are innovations in community settings. Our examples from different international research settings—New York City (NY, US), Madrid (Spain), and Santa Luzia and Mesquita (Rio de Janeiro, Brazil)—suggested that practitioner–researcher partnerships are a viable tool for translating research into practice in health service environments. Nonetheless, we are cognizant that there are challenges in comparing practitioners in these three different sociocultural and historical contexts. While all three contexts are of a medium to large size, urban, and densely populated cities with a relatively high volume of scientific research, practitioners in service agencies in Spain and Brazil, compared to those in the US, are much less involved in scientific research. This is because the US commits far greater financial/structural resources to research than other countries [37]. Brazil and Spain also lack resources or community capacity due to infrastructure issues. The post-military regime in Brazil (1964–1985) and Spain (1939–1975) left both of these countries depleted of resources and unable to develop a strong research agenda.

## 2. Practitioner–Researcher Partnerships

Based on our past research, herein we outline specific tasks and procedures (i.e., processes) that practitioners can perform in different phases of the research cycle. We offer recommendations for engaging practitioners in research partnerships, and for overcoming potential barriers to engaging practitioners in research. Our work is grounded in community-engaged research and implementation science principles and practices promoting and sustaining collaboration with practitioners [38]. Policymakers, funders, and professional organizations (e.g., the National Association of Social Workers and the American Public Health Association) recommend practitioner engagement in research, but have provided few guidelines for how to establish and maintain these partnerships [39,40,41,42,43]. International partnerships have even fewer guidelines, and research has enumerated a multitude of challenges [44,45]. Implementation research has been focused instead almost solely downstream on specific types of practitioners and their implementation of specific interventions in specific settings. For example, in the United States, practice-based research networks (PBRNs) engage only medical practitioners working in primary care practice settings [46]. Although PBRNs are important, all types of practitioners from a variety of service organizations—not only physicians and nurses in primary care settings—should be involved in research. Social justice is advanced through partnerships with organizations that serve communities most affected by health disparities and social inequities by enabling the development of practice and policy interventions that may be implemented by practitioners and policymakers [47]. Health and social outcomes are enhanced by the expertise of different practitioners in different settings, working across disciplines and using a collaborative-care approach [48]. For example, global efforts to prevent HIV transmission prioritize HIV testing and primary care [49]. Social and public health services providers—social workers, health educators, care navigators—are instrumental in linking at-risk patients to medical personnel licensed to perform HIV testing and deliver HIV primary care [50,51,52].

## 3. Factors that Influence Practitioner–Researcher Partnerships

Agency settings employ diffusion systems in which practitioners may share professional knowledge, clients, funding sources, areas of interest, and geographic communities. Diffusion requires adoption, implementation, and sustainment of innovations by practitioners. New practices, ideas, or technologies (“innovations”) often refer to interventions that are aimed at changing behavior or organizational structures, systems, or practitioner trainings. The practitioner–researcher partnership is therefore itself an innovation in that it represents a new practice that is aimed at improving services by accelerating the adoption of research-based findings in agency settings. Organizational culture and context, practitioner demographics, and client satisfaction all influence implementation of new practices [53]. “Implementation” is the action of introducing or adapting practices within community settings in such a way that they become routinely used and accepted as standard practice [54,55]. Thus, practitioner–researcher partnerships are likely to occur in organizations where the diffusion of innovations is already supported through organizational culture and social norms.

A culture favoring research is often present in well-resourced organizations with stable, functional environments marked by high job satisfaction, low job-related stress, and low turnover [56]. Organizations previously involved in research tend to have adequate office space, staff, and budgets to conduct program evaluations, needs assessments, and intervention studies [57]. Understaffed settings with limited resources may not have the capacity to participate in research [58]. Administrators may be reluctant to allow practitioners to work with researchers for fear of sacrificing patient contact or billable services [59]. Settings with limited office space or technology may be less able to engage in research without disrupting services [60,61].

While it is not possible for all practitioners to be involved in research at all times, organizations that participate in partnerships can foster social norms of positive regard for research among practitioners. Engaging practitioners in research requires developing informal and professional relationships through ongoing, candid, and frequent communication. Strong communication is best achieved through scheduled in-person meetings and, between meetings, by maintaining contact by telephone, email, Skype, and the like [62]. Partner engagement requires consistent attention to social cues—for example, social manners, timely responses to requests and emails, and overall flexibility about where and when meetings take place. Practitioners prioritize practice, and sometimes engaging in partnerships may fall out of the scope of their job descriptions. Therefore, ever-present time constraints must be met with a willingness to accommodate [63].

Researchers seeking to develop partnerships must demonstrate awareness of local issues and community concerns. Knowing the community’s history, norms, values, and health and social needs may help them build trust and gain consensus on priorities. Partnerships may advance equity among partners by seeking funding that is distributed to practitioners and researchers, although this can be challenging from a funding standpoint as well as from a fiscal management standpoint. Other challenges include a lack of trust due to a history of abuses by researchers studying vulnerable populations and the stigmatization of communities [64], researchers’ lack of funding to compensate community partners [65], imbalances of power and knowledge-sharing between researchers and their partners [66], and lack of sharing of research findings [67]. These barriers can be overcome by enhancing agency capacity—directly funding partnerships between communities and researchers conducting scientific research. In other words, by developing resources to share with agencies, including funding, staff, and space, agencies may be better positioned to participate in research partnerships, without sacrificing their missions of service provision [68].

## 4. Practitioners’ Various Roles in Research

Practitioners’ involvement in research can improve health and human services because such involvement enhances both practitioners’ skills and their adoption of research-based practices [69]. Nevertheless, the 2014 Public Health Workforce Interests and Needs Survey, perhaps the most comprehensive health workforce survey, lacks data that might describe what is needed in order for practitioners to have a role in research or to benefit from the advantages of practitioner involvement. To narrow this gap, based on our previous collaborative work with practitioners with a myriad of job descriptions and titles, we developed a summary of reasons why researchers ought to involve practitioners in social and health research. Table 1 shows that practitioners can be involved in each phase of the research cycle and suggests that their contributions vary, as do the benefits that they experience as a result of collaborative participation. We have highlighted the numerous types of opportunities for practitioners to offer consultation, expertise, and support, so that there is a menu of options for both researchers and practitioners depending on their resources, capacities, preferences, and research needs. This table demonstrates that practitioners’ contributions are grounded in their expertise, local knowledge, and their access to populations that researchers seek to engage, study, and develop interventions for. Likewise, researchers’ collaborative effort to retain partnerships with practitioners sensitizes practitioners to the benefits, limitations, and ultimate usefulness of research in their own work. This new knowledge that is created for both parties is unique and contributes depth and meaning to the work of providing services.

In partnerships, practitioners may serve as “cultural brokers,” negotiating and facilitating relationships between researchers and community members to help researchers gain access to and develop trust with local communities [70,71]. While several studies have referred to the involvement of practitioners in research [58,61,72,73,74,75], the scope of such “involvement” is seldom clarified. There is a dearth of empirical data showing specific research tasks and procedures with which practitioners are involved. Knowing these tasks and procedures could begin to fill the gap in the 2014 Public Health Workforce Interests and Needs Survey and point to a strategic plan that may be developed to guide future partnerships. Given practitioners’ limited time and the competing demands on their attention, it is prudent to specify roles and responsibilities, in advance, that are the most appropriate and desirable for practitioners. Therefore, to demonstrate the wide-ranging opportunities for involvement of practitioners in public health research, we highlighted the types of research tasks and procedures in previous community-engaged studies conducted by the authors in the United States, Spain, and Brazil. These studies involved diverse samples of practitioners (e.g., counselors, peer educators, physicians, nurses, and community-health workers) in New York City (140 practitioners in 24 community settings), Madrid (140 practitioners in 24 community settings), and Mesquita and Santa Luzia (168 community-health workers, 62 nurses, and 32 physicians) [69,76,77]. In these studies, the authors were guided by the principles of Community-Based Participatory Research [78], in creating and maintaining a collaborative board comprised of participants from HIV behavioral intervention research, researchers and practitioners that serve individuals affected by HIV. The collaborative board oversees, guides, and helps to develop all research proposals, grants, and study procedures. Research questions arise from the stated needs and interests of the board through an iterative process that fosters equity, accountability and transparency. This collaborative board approach of combining the expertise of different constituencies (e.g., practitioner, researcher, and participant or local resident) to develop research questions has been demonstrated to work internationally in Spain and for other health issues, for example obesity [79].

Table 2 shows that the lowest practitioner involvement in research was observed in Madrid, Mesquita, and Santa Luzia, while New York City had the highest involvement. Partnerships require that researchers allocate significant resources to recruit, train, supervise, and retain practitioners. Therefore, New York City, with its vast academic and service infrastructure, appears better equipped than the other cities in question. However, Brazil, with vastly fewer resources, showed demonstrably greater involvement in certain areas of research, including data collection, survey development and dissemination of findings, than Spain and in some cases NYC. This table gives an indication of the nuanced nature of involving practitioners based on the context, cultural norms around practitioner roles, available funding for training and compensation, and accepted practices for researchers. For example, in Brazil, researchers readily involve practitioners in data collection and less so in interventions. By viewing this table, we can draw some inferences about the research landscape and where gaps may exist that can be filled with greater collaboration. The lowest involvement in planning was observed in Madrid, and the greatest in Mesquita and Santa Luzia. Practitioners who participated in the aforementioned studies were also asked about their involvement in developing procedures for collecting data (such as interviews and surveys). In New York and in Madrid, data-collection procedures must be submitted before funding is obtained. Most of the planning work has been completed by the time practitioners join a team. Practitioners in Brazil appear to place a high value on the experiential, indigenous knowledge of practitioners about the use of language, terminology, and culturally sensitive ways to elicit information and use it to inform the development of interviews and procedures for collecting data. The lowest proportion of practitioners having ever been involved in research was observed in cities in Brazil, and the highest in New York. This is likely an artifact of the infrastructure already in place in New York City for research partnerships; Brazil has fewer resources and fewer formal mechanisms for supporting partnerships. Nevertheless, the highest proportion of practitioners interested in becoming involved in research was observed in Brazil. This may be indicative of an overall positive regard for and acceptance of the value of research in middle-income countries.

While there are many contextual differences that are worth noting between New York City, Brazil, and Spain, previous research has outlined the need to make comparisons across these countries that can inform development of strategies for engaging practitioners in research while honoring these differences [80]. As research becomes increasingly globalized and scientific information is presented internationally through all forms of media, adapting and exporting collaborative strategies is increasingly feasible.

## 5. Sustaining Successful Partnerships

Our work in different locations suggests that practitioners have both experience and an interest in research. However, challenges that hinder practitioners’ engagement in research partnerships remain. Partnerships are contingent not only on institutional support, but also on community-level support, interest, and commitment and on policies that allow partnerships to flourish [28,81]. Below, we discuss three key elements of community-engaged research that can help sustain practitioner–researcher partnerships.

(1) Negotiating common issues in partnerships: Conflicts arise between administrators in service organizations, practitioners, and researchers regarding many aspects of research (e.g., aims, design, sampling, recruitment and the collection, analysis, and interpretation of data) and for a variety of reasons. In order to foster mutual trust and support, partners ought to speak candidly about issues of power and seek to build consensus for moving forward. For example, agencies value program evaluations to help seek funding and improve services, while researchers value answering scientific questions [81]. Organizations may object to clients participating in research, because it may seem at odds with the provision of services. When such conflicts go unresolved or are not discussed, friction may follow, with organizations withdrawing their support and practitioners withdrawing their participation from partnerships [82,83]. Furthermore, practitioners may have mixed feelings about research due to a legacy of racism, racial- or gender-based violence, exploitation, or social harm at the hands of the research industry. Training practitioners in ethical principles of research—for example, justice and beneficence, grounded in the Belmont Report [84]—while training researchers in partnership formation may help resolve negative feelings and help maintain practitioners’ engagement [11,85,86].

(2) Disseminating research findings: As community-engaged research proliferates worldwide, demand is increasing for standardized methods, so that outcomes of partnerships may be replicated [87,88]. Community-engaged research has shown greater potential than traditional research paradigms for closing the research-practice gap by including practitioners on the research team and by facilitating dissemination of research findings [89]. Dissemination is perhaps the most essential role of practitioners who translate research into practice by implementing innovations. Practitioners share social support (material, informational, and emotional), attend training, and influence organizational culture and codes of conduct concerning ethical practices in order to provide effective services [4,5,6] (e.g., social and professional networks serve as natural outlets for dissemination of research-based practices) [90]. Thus, the literature suggests that practitioners are better able to bridge research and practice once they have been involved in research [18,20,21,22].

(3) Producing research-related outputs in partnership: The ways in which research partners integrate diverse knowledge and skill sets will differ from project to project and will evolve from initial engagement to dissemination. Partners can mitigate challenges by engaging in processes that have been shown to enhance partnership outcomes—power sharing, buy-in, collective problem solving, knowledge exchange, and capacity building [91]. These processes can be used to integrate diverse knowledge sets reflecting a balanced and coordinated distribution of roles, responsibilities, and tasks [92]. Practitioners’ involvement in research tasks and procedures is a unique outcome of participatory research; it might include manuscript writing, dissemination of findings at professional and academic conferences, and adaption and implementation of institutional review board protocols. Were such activities better understood, practitioners and researchers could make improvements to their work while strengthening the partnership [93]. For example, if practitioner involvement in research could be optimized through interventions aimed at developing, sustaining and cultivating partnerships over time, practitioners would be more likely to actually apply the findings of research to their daily work and to close the large gap between research and practice discussed previously [66,68,94].

## 6. Conclusions

As community-engaged research and implementation of research-based practices proliferate, practitioner–researcher partnerships will face challenges. Researchers favoring full control over the research may be unable to sustain partnerships. Cultural differences between the service and academic spheres may also impede partnerships. These differences may include timing (the “pace” of research is slower than that of service provision), jargon (the language of research does not reflect practice), and norms (researchers may be more socially “formal” than practitioners). These differences can be overcome by displaying cultural humility [27]—a stance grounded in dialectic processes combined to negotiate different interests and pursuits between researchers and practitioners, mutual support to overcome social and professional differences, and problem solving to help achieve consensus [69].

Given the vast potential for participatory partnerships to bridge the gap between research and practice, it is imperative that investments be made at the neighborhood, city, state, and national levels to foster inclusion of practitioners in the development of priorities that might affect social and public health services. In order to advance research, policy, and practice concerning practitioner–researcher partnerships, we suggest the following next steps:

(1) Future research ought to examine how practitioner involvement in research tasks may influence practitioners’ commitment to translating research into practice. Knowing how to motivate practitioners to use scientific evidence to guide practice will help to engage practitioners. Therefore, resources for enhancing partnership–management skills ought to be developed for both researchers and practitioners.

(2) Future involvement of practitioners in research will depend on how they regard the need for scientific research. Training practitioners and administrators in research ethics and practice can help. Such training ought to focus on messages salient to practitioners and remain jargon-free. Partnerships should share responsibility for translating research into practice and disseminating “lessons learned” that could be applied in a variety of contexts.

(3) Future publications need to specify the exact nature of practitioner involvement, so that partnerships may better understand and replicate the steps taken to advance the aspects of research emphasized above: Engagement, methods, dissemination, and evaluation.

(4) Future state and local policies can help partnerships overcome key interrelated challenges, such as lack of time and financial resources. Funding agencies can require researchers and practitioners to share budgetary resources. Financial equity between practitioners and researchers may also facilitate partnership building.

Practitioners advocate for community members and promote their health and social well-being while working to prevent illness and conditions that cause poor health. Practitioners involved in research endorse research-based practices that have the greatest potential to create the strongest impact. In general, these practitioners are more aware than their peers of the importance of integrating research knowledge with practice wisdom. Therefore, we recommend practitioner–researcher partnerships in order to enhance the usefulness of research, to guide the methods of culturally responsive research, and to enhance the implementation of research-based services and programs.

## Figures and Tables

**Table 1 ijerph-16-00862-t001:** Advantages and opportunities for practitioner–researcher partnerships.

Phase of Research	Advantages of Involving Practitioners in Research	Opportunities for Practitioners
Engagement	▪ Develop leadership ▪ Help researchers understand local issues ▪ Build consensus ▪ Introduce local theories ▪ Write grant applications ▪ Distribute tasks and procedures	▪ Share power ▪ Encourage buy-in ▪ Facilitate dissemination ▪ Solve problems ▪ Exchange knowledge ▪ Build capacity
Methods	▪ Define methods ▪ Identify, select, and refine measures ▪ Represent local theories ▪ Identify existing local interventions ▪ Translate and adapt interventions ▪ Screen participants ▪ Act as research assistants ▪ Manage and code data ▪ Analyze and interpret data	▪ Improve relevance of research aims ▪ Improve comprehensibility of measures ▪ Help scientific interventions resemble natural local interventions ▪ Bridge research and practice by adopting and delivering evidence-based practices ▪ Add to practice wisdom
Dissemination	▪ Write and review papers ▪ Disseminate reports ▪ Choose outlets for publication	▪ Improve dissemination of findings by diversifying outlets ▪ Expand meaning of findings
Evaluation	▪ Identify local politics and concerns ▪ Reflect practice wisdom ▪ Represent clients’ voices	▪ Share power and solve problems ▪ Exchange knowledge and encourage buy-in ▪ Build capacity
Implementation	▪ Deliver EBIs ▪ Manage and maintain EBIs ▪ Sustain champions of EBIs ▪ Translate and modify EBIs to adhere to local cultures and norms	▪ Maintain fidelity and effectiveness of established programs ▪ Prevent unintended effects when programs are transferred from labs to community and other settings ▪ Integrate multiple interventions for better cost-effectiveness

**Table 2 ijerph-16-00862-t002:** Practitioner involvement in social and health research in different contexts.

Practitioner Involvement	Phase of Research	Brazil %	Spain %	US %
Procedural Involvement	Data Collection	45	27	67
	Intervention Facilitation	24	21	50
	Participant Recruitment	24	25	62
	Participant Interviewing	21	23	62
Substantive Involvement	Survey Development	32	15	45
	Data-Collection Procedures	77	14	42
	Data Analysis	16	15	30
	Dissemination of Findings	33	10	35
Previous Involvement	Have you participated in research?	15	40	89
Involvement Intention	Would you like to be involved in research?	94	71	88

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
