# Peer review of "Nurturing Practitioner-Researcher Partnerships to Improve Adoption and Delivery of Research-Based Social and Public Health Services Worldwide"

_ijerph, 2019, doi:10.3390/ijerph16050862_

Reviewer 1 Report

It needs to reduce the size of the article and instead introduce the tables with better explanation and discussion. There is limited discussion of the table 1 and particularly table 2. Also, this study should outline upfront the challenges of comparing NY, Spain and Brazil considering the lack of resources or community capacity and infrastructure issues. Overall post-military authoritarianism in Brazil, we do see greater interest in social participation in decision-making. Is this true in Spain or socio-cultural contexts needs to be explained. Therefore, the theoretical framework is nicely presented but the context of each case remains weak. 

Author Response

Reviewer 1

We thank the reviewers for the helpful comments and in particular that the theoretical framework was a key strength of this manuscript. Below, we address each comment in italics with our response and corresponding changes in the text (highlighted in green throughout the main document).

Comment #1: It needs to reduce the size of the article and instead introduce the tables with better explanation and discussion. There is limited discussion of the table 1 and particularly table 2.

We agree and with respect to Table 1, we have added discussion on lines 183-188. For Table 2, we have added discussion on lines 219-226. Also, we have provided some additional discussion of the international context challenges on lines 240-245.

We respectfully decline to reduce the article because we fell we would need details that are important for an international audience.

Comment #2: Also, this study should outline upfront the challenges of comparing NY, Spain and Brazil considering the lack of resources or community capacity and infrastructure issues. Overall post-military authoritarianism in Brazil, we do see greater interest in social participation in decision-making. Is this true in Spain or socio-cultural contexts needs to be explained.

In lines 96-104, we outlined the differences across the three contexts and offer information about challenges in comparing them, e.g., lack of resources in Brazil and Spain. We also mentioned the dictatorship in those countries. We cannot go beyond this because the data we have are not conclusive to explain socio-cultural contexts. This is not the aim of this paper.

Reviewer 2 Report

The matter of the manuscript is interesting. It has relevance for reflection on practice and research in the social community.  However, the manuscript has some weakness, which must be improve.

General comments: The yellow underline has to be removed from a presentation of an original document. The objective of the paper and methodology must be clarified. The novelty of the work must be highlighted and clarified.

If the paper is an “opinion paper” it has to uses the most recent references, in order to show the current situation. The intention of the authors is to demonstrate the importance of including workers and researchers in the investigations and initiate a collaboration between both. Referenced studies, with more recent dates, would be necessary. I recommend introducing updated references, around year 2019.  At the same time, it is important to reference other studies with a different or opposite point of view. All the referenced studies are in the same direction but it is not very common in research.  It is important in order to avoid the social desirability bias and make an open-ended discussion. Could you reference other international studies, with opposite perspectives or with alternative results around the author’s point of view?

There are some comments regarding the structure of the paper: It starts with the number 1 in “Introduction” and never see again this kind of numerical classification. “Sustaining successful partnerships” paragraph could be better structured, (maybe change the underline and introduce enumeration?).

Other Specific comments:

p.94. A more specific description of the objective of the paper should be great before starting the “Practitioner-Researcher Partnerships” paragraph.

p.135. We are in 2019. What happened in the last 4 years? References must be updated “Organizations previously involved in research tend to have adequate office space, staff, and  budgets to conduct program evaluations, needs assessments, and intervention studies (Joe, Broome, Simpson, & Rowan-Szal, 2007). Understaffed settings with limited resources may not have the capacity 137 to participate in research (Aarons et al., 2012; Glisson, 2007). Administrators may be reluctant to allow practitioners to work with researchers for fear of sacrificing patient contact or billable services (Brown, Wickline, Ecoff, & Glaser, 2009; Wells et al., 2006). Settings with limited office space or technology may  be less able to engage in research without disrupting services (Pagoto et al., 2007).

p.145 What scientific evidence do you have about this affirmation? Try to use scientific references to support that idea. “Strong communication is best achieved through scheduled in-person meetings and, between meetings, by maintaining contact by telephone, email, Skype, and the like”.

p.156-161. We are in 2019. What happened in the last 4 years? References must be updated. “Other challenges include a lack of trust due to a history of abuses by researchers studying vulnerable populations and the 157 stigmatization of communities (Corbie-Smith, Thomas, & St. George, 2002; Rencher & Wolf, 2013), researchers’ lack of funding to compensate community partners (Lantz, Israel, Schulz, & Reyes, 2005), imbalances of power and knowledge-sharing between researchers and their partners (Mason & Boutilier, 160 2009), and lack of sharing of research findings (Cashman et al., 2008)

p163. I agree and I suppose it is a desire of the authors, but in order to conclude it, we would need more scientific support (recent studies): In other words, by developing resources to share with agencies, including funding, staff, and space, agencies may be better positioned to participate in research partnerships, without sacrificing their missions of service provision”

174. is it the objective of the paper? ”we developed a summary of reasons why researchers ought to involve practitioners in social and health research”.

p.184. Reference those several studies “While several studies have referred to the involvement of practitioners in research”

p206. It is not included in references “(Díez et al., 2018).”

p238. It is interesting and well referenced For example, agencies value program evaluations to help seek funding and improve services, while researchers value answering scientific questions (Thompson et al., 2009).”

p.275. It is really interesting, but do you have scientific references or experiences published to argue it? “For example, if practitioner involvement in research could be optimized through interventions aimed at developing, sustaining and cultivating partnerships over time, practitioners would be more likely to actually apply the findings of research to their daily work and to close the large gap between research and practice discussed previously”.

p.296 You are right, how can we proceed with that? What do you propose? “However, without further empirical data, the conceptualization provided here is not sufficient to explain fully the impact of practitioner involvement in research, or practitioners’ actual use of research findings, or the impact such involvement may have on the lives of clients”

p.302-307. This sentence is exactly the same in two different spaces in the paper. I seriously recommend reviewing the paper and remove duplicities. “These differences may include timing (the “pace” of research is slower than that of service provision); jargon (the language of research does not reflect practice); and norms (researchers may be more socially “formal” than practitioners)….”.

p332. How do you reach this conclusion? It is important that there is a clear relationship between the scientific results and the proposals. Please, clarify it.  In general, these practitioners are more aware than their peers of the importance of integrating research knowledge with practice wisdom.”

Author Response

Reviewer 2

We thank the reviewer for noting that, “The matter of the manuscript is interesting. It has relevance for reflection on practice and research in the social community.”  

Comment #1: The yellow underline has to be removed from a presentation of an original document. The objective of the paper and methodology must be clarified. The novelty of the work must be highlighted and clarified.

We have removed the yellow highlighting. All changes are highlighted in green.

We have added text to line 93-107 to respond to the recommendation for clarification of objective and novelty of the work. Our methodology is highlighted yet we would like to respectfully point out that this is not a research paper per se. We were advised as such by the editor and we resubmitted this manuscript as an “opinion piece” to follow his astute recommendation as he noted that we did not need a traditional “methods” section. Our paper uses data in order to illustrate the points we are making, which have stated on line 103.

Comment #2 If the paper is an “opinion paper” it has to uses the most recent references, in order to show the current situation. The intention of the authors is to demonstrate the importance of including workers and researchers in the investigations and initiate a collaboration between both. Referenced studies, with more recent dates, would be necessary. I recommend introducing updated references, around year 2019.  At the same time, it is important to reference other studies with a different or opposite point of view. All the referenced studies are in the same direction but it is not very common in research.  It is important in order to avoid the social desirability bias and make an open-ended discussion. Could you reference other international studies, with opposite perspectives or with alternative results around the author’s point of view?

We respectfully disagree with the reviewer’s position that an “opinion piece” must provide contrary viewpoints. Such stance would better match the format of a “commentary.” Nonetheless, we have provided evidence in the form of data and literature to support the view that collaboration between practitioners and researchers can and do lead to better outcomes. We are not aware of any papers or commentaries relating evidence that collaboration leads to worse outcomes. Nonetheless, we have followed this reviewer’s recommendation and have sprinkled more recent references throughout the paper in order to further strengthen our position.

Comment #3: There are some comments regarding the structure of the paper: It starts with the number 1 in “Introduction” and never see again this kind of numerical classification. “Sustaining successful partnerships” paragraph could be better structured, (maybe change the underline and introduce enumeration?).

We have added enumeration as recommended for improved structure and removed the 1 from the Introduction heading.

Other Specific comments:

p.94. A more specific description of the objective of the paper should be great before starting the “Practitioner-Researcher Partnerships” paragraph.

We have added a more specific description of the objective of the paper to lines 107-112 as suggested by the reviewer.

p.135. We are in 2019. What happened in the last 4 years? References must be updatedOrganizations previously involved in research tend to have adequate office space, staff, and  budgets to conduct program evaluations, needs assessments, and intervention studies (Joe, Broome, Simpson, & Rowan-Szal, 2007). Understaffed settings with limited resources may not have the capacity 137 to participate in research (Aarons et al., 2012; Glisson, 2007). Administrators may be reluctant to allow practitioners to work with researchers for fear of sacrificing patient contact or billable services (Brown, Wickline, Ecoff, & Glaser, 2009; Wells et al., 2006). Settings with limited office space or technology may  be less able to engage in research without disrupting services (Pagoto et al., 2007).

We have added the following updated references to the text as suggested:

Hamilton, A. B., Mittman, B. S., Campbell, D., Hutchinson, C., Liu, H., Moss, N. J., & Wyatt, G. E. (2018). Understanding the impact of external context on community-based implementation of an evidence-based HIV risk reduction intervention. BMC Health Services Research, 18(1), 11. doi:doi.org/10.1186/s12913-017-2791-1

McBeath, B., & Austin, M. J. (2015). The organizational context of research-minded practitioners: Challenges and opportunities. Research on Social Work Practice, 25(4), 446-459.

Despard, M. R. (2016). Challenges in implementing evidence-based practices and programs in nonprofit human service organizations. Journal of evidence-informed social work, 13(6), 505-522.

Spector, A. Y., & Pinto, R. M. (2017). Partnership matters in health services research: A mixed methods study of practitioners’ involvement in research and subsequent use of evidence-based interventions. Journal of Mixed Methods Research, 11(3), 374-393.

Beidas, R. S., Stewart, R. E., Adams, D. R., Fernandez, T., Lustbader, S., Powell, B. J., . . . Hurford, M. O. (2016). A multi-level examination of stakeholder perspectives of implementation of evidence-based practices in a large urban publicly-funded mental health system. Administration and Policy in Mental Health and Mental Health Services Research, 43(6), 893-908.

p.145 What scientific evidence do you have about this affirmation? Try to use scientific references to support that idea. “Strong communication is best achieved through scheduled in-person meetings and, between meetings, by maintaining contact by telephone, email, Skype, and the like”.

We agree and have added the following reference after the above sentence on line 159: Pinto, R.M, Wall, M., Spector, A.Y., (2013). Modeling the structure of partnership between researchers and front-line service providers: Strengthening collaborative public health research. Journal of Mixed Methods Research, doi: 10.1177/1558689813490835

p.156-161. We are in 2019. What happened in the last 4 years? References must be updated. “Other challenges include a lack of trust due to a history of abuses by researchers studying vulnerable populations and the 157 stigmatization of communities (Corbie-Smith, Thomas, & St. George, 2002; Rencher & Wolf, 2013), researchers’ lack of funding to compensate community partners (Lantz, Israel, Schulz, & Reyes, 2005), imbalances of power and knowledge-sharing between researchers and their partners (Mason & Boutilier, 160 2009), and lack of sharing of research findings (Cashman et al., 2008)

We have added the following updated references:

George, S., Duran, N., & Norris, K. (2014). A systematic review of barriers and facilitators to minority research participation among African Americans, Latinos, Asian Americans, and Pacific Islanders. American journal of public health, 104(2), e16-e31.

Ion, G., Stîngu, M., & Marin, E. (2018). How can researchers facilitate the utilisation of research by policy-makers and practitioners in education? Research Papers in Education, 1-16. doi:10.1080/02671522.2018.1452965

Samuel, C. A., Lightfoot, A. F., Schaal, J., Yongue, C., Black, K., Ellis, K., . . . Foley, K. (2018). Establishing new community-based participatory research partnerships using the community-based participatory research Charrette model: Lessons from the Cancer Health Accountability for Managing Pain and Symptoms Study. Progress in community health partnerships: Research, education, and action, 12(1), 89-99.

Langley, J., Wolstenholme, D., & Cooke, J. (2018). ‘Collective making’as knowledge mobilisation: The contribution of participatory design in the co-creation of knowledge in healthcare. BMC Health Services Research, 18(1), 585. doi:10.1186/s12913-018-3397-y

p163. I agree and I suppose it is a desire of the authors, but in order to conclude it, we would need more scientific support (recent studies): “In other words, by developing resources to share with agencies, including funding, staff, and space, agencies may be better positioned to participate in research partnerships, without sacrificing their missions of service provision”

We agree and have added an updated reference on line 178.

174. is it the objective of the paper? ”we developed a summary of reasons why researchers ought to involve practitioners in social and health research”.

This summary leads us into our recommendations, which are the main objective as stated on line 80.

p.184. Reference those several studies “While several studies have referred to the involvement of practitioners in research”

We have added the following references:

Pinto, R. M., Witte, S. S., Wall, M. M., & Filippone, P. L. (2018). Recruiting and retaining service agencies and public health providers in longitudinal studies: Implications for community-engaged implementation research. Methodological Innovations, 11(1-13). doi:10.1177/2059799118770996

Spector, A. Y., & Pinto, R. M. (2017). Partnership matters in health services research: A mixed methods study of practitioners’ involvement in research and subsequent use of evidence-based interventions. Journal of Mixed Methods Research, 11(3), 374-393.

Beidas, R., Skriner, L., Adams, D., Wolk, C. B., Stewart, R. E., Becker-Haimes, E., . . . Weaver, S. (2017). The relationship between consumer, clinician, and organizational characteristics and use of evidence-based and non-evidence-based therapy strategies in a public mental health system. Behaviour Research and Therapy, 99, 1-10. doi:10.1016/j.brat.2017.08.011

Hohl, S. D., Thompson, B., Krok-Schoen, J. L., Weier, R. C., Martin, M., Bone, L., . . . Calderón, N. E. (2016). Characterizing community health workers on research teams: results from the Centers for Population Health and Health Disparities. American Journal of Public Health, 106(4), 664-670.

Despard, M. R. (2016). Challenges in implementing evidence-based practices and programs in nonprofit human service organizations. Journal of Evidence-Informed Social Work, 13(6), 505-522.

Rachmawati, K., Schultz, T., & Cusack, L. (2017). Translation, adaptation and psychometric testing of a tool for measuring nurses’ attitudes towards research in Indonesian primary health care. Nursing Open, 4(2), 96-107.

p206. It is not included in references “(Díez et al., 2018).”

We have added this.

p.275. It is really interesting, but do you have scientific references or experiences published to argue it? “For example, if practitioner involvement in research could be optimized through interventions aimed at developing, sustaining and cultivating partnerships over time, practitioners would be more likely to actually apply the findings of research to their daily work and to close the large gap between research and practice discussed previously

We have added the following references:

Bowen, S., Botting, I., Graham, I. D., & Huebner, L.-A. (2017). Beyond" two cultures": Guidance for establishing effective researcher/health system partnerships. International Journal of Health Policy and Management, 6(1), 27-42. doi:10.15171/ijhpm.2016.71

Langley, J., Wolstenholme, D., & Cooke, J. (2018). ‘Collective making’as knowledge mobilisation: The contribution of participatory design in the co-creation of knowledge in healthcare. BMC Health Services Research, 18(1), 585. doi:10.1186/s12913-018-3397-y

M., Karltun, J., Keller, C., & Gäre, B. A. (2018). Collaborative and partnership research for improvement of health and social services: Researcher’s experiences from 20 projects. Health Research Policy and Systems, 16(1), 46. doi:https://doi.org/10.1186/s12961-018-0322-0

p.296 You are right, how can we proceed with that? What do you propose? “However, without further empirical data, the conceptualization provided here is not sufficient to explain fully the impact of practitioner involvement in research, or practitioners’ actual use of research findings, or the impact such involvement may have on the lives of clients”

We appreciate the reviewer’s comment, but wish to state that the goal of this paper was not to provide the answers to all questions we posed. We aimed to illuminate practical issues that can improve collaboration between researchers and practitioners. In no place in the text do we say that this paper has the ultimate answers to the issues we posed. Indeed, we have made it clear that this paper is intended to provide recommendations (see introduction) that can be used by research and practitioners in different contexts. We therefore propose, as explained throughout the text, recommendations that can be used to improve collaboration between practitioners and researchers.

p.302-307. This sentence is exactly the same in two different spaces in the paper. I seriously recommend reviewing the paper and remove duplicities. “These differences may include timing (the “pace” of research is slower than that of service provision); jargon (the language of research does not reflect practice); and norms (researchers may be more socially “formal” than practitioners)….”.

We thank the reviewer and have removed the text from the introduction from line 79, where it first appeared and it now resides in the conclusion section, line 302, where it was intended.

p332. How do you reach this conclusion? It is important that there is a clear relationship between the scientific results and the proposals. Please, clarify it.  In general, these practitioners are more aware than their peers of the importance of integrating research knowledge with practice wisdom.”

We agree and have added a reference on line 354 that supports this conclusion. 

Round  2

Reviewer 1 Report

With changes made and justifications made, I am satisfied.